# Partial Physics Informed Diffusion Model for Ocean Chlorophyll Concentration Reconstruction

**Qianxun Xu**
Division of Natural and Applied Sciences
Duke Kunshan University
Kunshan, China
qianxun.xu@dukekunshan.edu.cn

**Zuchuan Li**[*]
Division of Natural and Applied Sciences
Duke Kunshan University
Kunshan, China
zuchuan.li@dukekunshan.edu.cn

## Abstract

The integration of big data, physical laws, and machine learning algorithms has shown potential to improve the estimation and understanding of complex real-world systems. However, effectively incorporating physical laws with uncertainties into machine learning algorithms remains understudied. In this work, we bridge this gap by developing the Partial Physics Informed Diffusion Model (PPIDM), a novel framework that integrates known physical principles through a physics operator while reducing the impact of unknown dynamics by minimizing related discrepancies. We showcase PPIDM's capabilities using ocean surface chlorophyll concentration data, which are influenced by both physical and biological processes, while the latter is poorly constrained. Experimental results reveal that PPIDM achieves substantially improved prediction accuracy and stability, significantly outperforming baseline methods that either neglect physics entirely or impose incomplete physical constraints under the assumption of completeness. Code will be available here.

## 1 Introduction

Diffusion models generate samples from an unknown data distribution by reversing a forward noising process applied to clean data. They have demonstrated remarkable success in generating complex textures, structures, and motion patterns across a wide range of applications, excelling in generative tasks such as image synthesis [4, 7], video generation [6, 9, 8], and medical analysis [20, 22].

Despite these advances in generating content that is coherent and closely aligned with the underlying data distribution, diffusion models still face challenges when applied to scenarios where the generated data must strictly adhere to specific constraints. This is particularly evident in scientific and engineering applications, where the generated data must not only mimic real-world examples but also meet strict specifications and comply with fundamental physical laws. However, training a diffusion model on a dataset that meets specific constraints does not inherently ensure that the generated samples will strictly conform to those same constraints. As a result, incorporating explicit domain knowledge is essential for guiding the model toward a more sophisticated understanding of the data distribution and its underlying physical principles. Recent studies [1, 2, 5, 11, 17, 18] have made notable progress in this regard by embedding physical constraints directly into the models. These approaches typically assume that data can be fully characterized by well-defined constraints. In practice, however, these constraints are frequently incomplete, either due to the absence of critical parameter values or because of limited understanding of underlying processes. Consequently, existing methods such as minimizing the residual to zero [1, 17] or constraining the generated data within predefined bounds [2] may not guarantee physically consistent generations.

---

[*]Corresponding author.

To address these challenges, we propose the Partial Physics Informed Diffusion Model (PPIDM), which captures the reliable portions of the governing laws through a physics operator yet remains flexible to unmodeled or uncertain system components. PPIDM applies this operator to both real and generated data and penalizes discrepancies between their outputs, while allowing unmodeled or uncertain components of the system to be learned from data. This approach bridges the gap between theoretical constraints and data-driven adaptability, resulting in more accurate and physically consistent generations without compromising the model's ability to capture unknown or partially understood dynamics.

We demonstrate the performance of PPIDM on the reconstruction of oceanic chlorophyll (Chl) concentration, a task that involves infilling temporal gaps and predicting future values. Chl is a widely used proxy for phytoplankton biomass and oceanic primary productivity. Accurate reconstruction of Chl is crucial for understanding oceanic biogeochemical processes, monitoring ecosystem health, and assessing the ocean's response to climate change. Chl dynamics is governed by both physical and biological processes, and is often modeled using the Advection-Diffusion-Reaction partial differential equation (ADR PDE). The advection term of ADR can be well constrained using velocity fields, while the diffusion and reaction terms are more difficult to observe. In particular, the reaction term, encapsulating biological processes such as phytoplankton growth, depends on complex factors such as nutrient availability, light conditions, and community composition which are often intractable. Therefore, reconstructing models solely constrained by advection are incomplete and inaccurate. To address this, PPIDM integrates partial physical knowledge with data-driven learning. Our experimental results on ocean Chl data [21] demonstrate that PPIDM effectively balances domain knowledge with observational data, outperforming purely data-driven baselines and naive implementations that assume complete physics.

## 2 Related Works

### 2.1 Denoising Diffusion Probabilistic Models

Diffusion models are a class of probabilistic generative models that learn to map samples from the true data distribution $q(\boldsymbol{x})$ into pure noise via a forward noising process, and then learn to invert this process to recover data from noise using a learned model distribution $p_\theta(\boldsymbol{x})$ [7, 19, 20].

Specifically, the forward diffusion process introduces Gaussian noise progressively to an initial data point $\mathbf{x}_0$ through a Markov chain across $T$ discrete steps. At each timestep $t$, noise is injected according to:

$$q(\mathbf{x}_t \mid \mathbf{x}_{t-1}) = \mathcal{N}(\mathbf{x}_t; \sqrt{1 - \beta_t}\mathbf{x}_{t-1}, \beta_t\mathbf{I}), \tag{1}$$

where the noise level is controlled by a pre-defined variance schedule $\beta_t \in [0, 1]$. Due to the Gaussian nature of each incremental step, the marginal distribution $q(\mathbf{x}_t \mid \mathbf{x}_0)$ at any timestep can be derived in closed form:

$$q(\mathbf{x}_t \mid \mathbf{x}_0) = \mathcal{N}(\mathbf{x}_t; \sqrt{\bar{\alpha}_t}\mathbf{x}_0, (1 - \bar{\alpha}_t)\mathbf{I}), \tag{2}$$

where $\alpha_t = 1 - \beta_t$ and $\bar{\alpha}_t = \prod_{s=1}^{t} \alpha_s$. As $t \to T$, the distribution converges toward a standard Gaussian $\mathcal{N}(\mathbf{0}, \mathbf{I})$, making $\mathbf{x}_T$ essentially pure noise.

The core challenge for diffusion models lies in reversing this noising process to generate samples from the data distribution $q(\mathbf{x}_0)$. Ideally, we would sample directly from the true posterior distributions $q(\mathbf{x}_{t-1} \mid \mathbf{x}_t)$. However, since these distributions are analytically intractable, diffusion models approximate them using parameterized conditional distributions $p_\theta(\mathbf{x}_{t-1} \mid \mathbf{x}_t)$ modeled by neural networks:

$$p_\theta(\mathbf{x}_{0:T}) = p(\mathbf{x}_T) \prod_{t=1}^{T} p_\theta(\mathbf{x}_{t-1} \mid \mathbf{x}_t), \tag{3}$$

where each conditional is defined as a Gaussian distribution parameterized by learned functions $\mu_\theta(\mathbf{x}_t, t)$ and $\Sigma_\theta(\mathbf{x}_t, t)$. Typically, $p(\mathbf{x}_T) \approx \mathcal{N}(\mathbf{0}, \mathbf{I})$, enabling an iterative denoising from noise to the original data. Training maximizes a variational bound on $\log p_\theta(\mathbf{x}_0)$; under the usual formulation this reduces to the noise prediction objective [7]:

$$L_t = \mathbb{E}_{\mathbf{x}_0, \boldsymbol{\epsilon}_t} \left[ \left\| \boldsymbol{\epsilon}_t - \boldsymbol{\epsilon}_\theta\big(\mathbf{x}_t(\mathbf{x}_0, \boldsymbol{\epsilon}_t), t\big) \right\|^2 \right], \tag{4}$$

where $\epsilon_t \sim \mathcal{N}(\mathbf{0}, \mathbf{I})$. This loss is equivalent to predicting the clean sample,

$$\hat{\mathbf{x}}_0 = \frac{1}{\sqrt{\bar{\alpha}_t}}\Big(\mathbf{x}_t - \sqrt{1 - \bar{\alpha}_t}\,\epsilon_\theta(\mathbf{x}_t, t)\Big). \tag{5}$$

In this work, we explicitly frame our objective in terms of predicting the clean signal $\mathbf{x}_0$, because this allows for more straightforward integration of physical constraints on the reconstructed state.

## 2.2 Physics Informed Machine Learning

Diffusion models have recently been extended to incorporate physical knowledge for scientific modeling. One prominent line of work enforces governing equations directly during training or sampling. For example, CoCoGen [11] incorporates discretized PDE constraints into the reverse diffusion process to ensure physically plausible generation. Similarly, PIDM [1] introduce PDE residual losses into the training objective to align generated samples with known physical laws. These models offer high fidelity under well-specified physics, but assume the governing equations are both complete and accurate, which often breaks down in real-world systems. Other methods condition the diffusion process on physics-derived signals. Projected Diffusion projects the state at every step onto constraint consistent manifolds that encode physical feasibility [2], whereas DiffusionPDE learns a joint distribution over coefficients and solutions and performs inference with sparse observations and physics guided updates [10]. Though effective, these models rely on fully known governing equations to define constraints or training distributions, and are less suited to settings with incomplete knowledge.

Beyond diffusion-based approaches, physics-informed machine learning frameworks [13, 14, 15, 16, 17] address partial knowledge by estimating unknown parameters or learning solution operators, then simulating forward from initial or boundary conditions. Although these methods accommodate inverse settings and incomplete physics, their primary objective is PDE identification or operator learning, and they typically yield a single forward trajectory per initial state. Incorporating irregular, multi-time conditioning at inference such as conditioning on arbitrary subsets of observed frames and generating ensembles of reconstructions typically requires additional data assimilation or explicit stochastic modeling beyond the base frameworks. This requirement limits the flexibility of these methods in real-world scenarios characterized by irregular observations.

In this work, we study diffusion models in which the governing physics is only partially known. Instead of requiring full equations or simulators, PPIDM introduces a physics operator $\phi$ that encodes only the trusted components of the dynamics and couples this partial theory with data-driven denoising through a physics residual difference. This design extends physics-informed diffusion to under-specified scientific systems and leverages the generative nature of diffusion to condition on any subset of observations, enabling the generation of multiple physically plausible reconstructions without retraining.

## 3 Partial Physics Informed Diffusion Model (PPIDM)

### 3.1 Problem Formulation and Physics Operator Construction

We consider a physical system whose true dynamics stem from both known physics and unobserved biological processes. Concretely, the state variable $\mathbf{x}(t)$ evolves according to:

$$\frac{\partial \mathbf{x}}{\partial t} = \mathcal{P}_{\text{known}}(\mathbf{x}, \mathbf{v}) + \mathcal{B}_{\text{unknown}}(\mathbf{x}), \tag{6}$$

where $\mathcal{P}_{\text{known}}$ models known processes such as advection with velocity field $\mathbf{v}$, and $\mathcal{B}_{\text{unknown}}$ encapsulates latent biological effects or other unresolved physics. Our main goal is to reconstruct the Chl concentration state $\mathbf{x}_0$ while preserving consistency with $\mathcal{P}_{\text{known}}$. The core challenge lies in enforcing partial physical knowledge without over-constraining the model where full dynamics remain unknown. Therefore, we define a physics-informed operator:

$$\phi : \mathcal{X} \to \mathcal{Y},$$

where $\mathcal{X}$ denotes the space of original states (*e.g.*, concentration fields or other physical quantities to be generated) and $\mathcal{Y}$ represents the space of physics-informed projections. This operator $\phi$ is

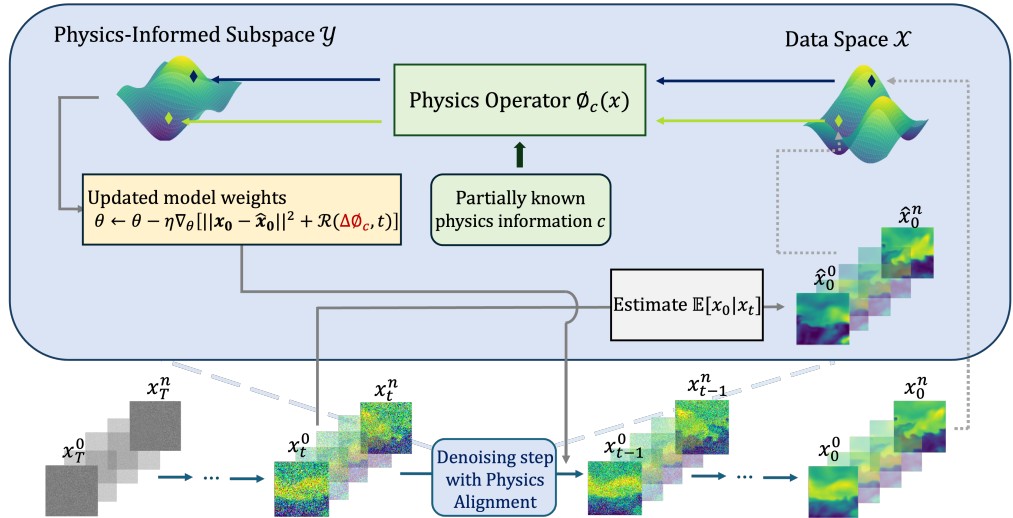

Figure 1: Overview of our proposed PPIDM: At each denoising step, the model projects predicted and ground truth clean signal onto the physics-informed subspace and updates model weights with the auxiliary physics loss term, guiding the model to learn the partially known physics.

constructed so that applying $\phi$ to a ground-truth or predicted states enforce and reveal consistency with the known physical laws. Specifically, for any state $\mathbf{x}_0$, $\phi(\mathbf{x}_0)$ can be viewed as its projection into a physics-informed subspace, which is the set of states that are consistent with the known but incomplete dynamics, or how the system would look if only the known physics governed it. The exact form of $\phi$ depends on the level and nature of the available physics knowledge:

**Partially Known Parameters:** When the governing equation is known in principle but contains unknown parameters, we selectively remove or omit terms involving those unknowns. We then build $\phi$ by applying the remaining or known part of the equation.

**Fully Known Subsystem:** When a law perfectly describes a partial subsystem and all associated parameters are certain, $\phi$ is defined to enforce this subsystem exactly. Even though the law itself is fully accurate for its domain, it does not address the rest of the system's dynamics. By applying $\phi$ to each sample, we ensure that known subsystems are satisfied, leaving the unknown effects such as additional or more complex processes to be learned from data.

### 3.2 Mechanism of the Physics Operator

Let $\mathbf{x}_0$ denote a ground truth sample from the data distribution and $\hat{\mathbf{x}}_0$ the corresponding model-generated sample at each step t. In a vanilla diffusion model, one typically minimizes a loss directly between $\mathbf{x}_0$ and $\hat{\mathbf{x}}_0$:

$$\mathcal{L}_{\text{data}} = \mathbb{E}_{\mathbf{x}_0,\epsilon,t} \left[ w(t) \| \mathbf{x}_0 - \hat{\mathbf{x}}_0(\mathbf{x}_t, t) \|^2 \right], \tag{7}$$

where $w(t) = \frac{1-\alpha_t}{1-\bar{\alpha}_t}$ is the weighting function and $\hat{\mathbf{x}}_0(\mathbf{x}_t, t)$ is the estimate of original data at each time step of reverse process utilizing Eq. 5. However, this loss is purely data-driven and does not incorporate domain knowledge. To leverage the physics operator $\phi$, unlike prior work that enforces $\phi(\hat{\mathbf{x}}_0) = 0$ based on fully known physical laws [1], we recognize that our system evolves under both $\mathcal{P}_{\text{known}}$ and $\mathcal{B}_{\text{unknown}}$. As such, the true state $\mathbf{x}_0$ itself does not strictly satisfy $\phi(\mathbf{x}_0) = 0$, so it is unreasonable to force the predicted state to satisfy known physics absolutely. Instead, we design the physics loss to encourage $\hat{\mathbf{x}}_0$ to satisfy the known physics to the same extent as the true state $\mathbf{x}_0$ does. Formally, we define a physics residual difference that quantifies the discrepancy between the projection of the predicted state and that of the ground truth:

$$\Delta\phi = \mathbb{E}_{t,\mathbf{x}_0} \left[ \phi(\mathbf{x}_0) - \phi(\hat{\mathbf{x}}_0(\mathbf{x}_t, t)) \right], \tag{8}$$

To avoid over-constraining the model at early timesteps when the predicted states are still highly uncertain, we adopt a progressively enforced constraint following the probabilistic formulation of [1].

Specifically, we interpret the residual difference $\Delta\phi$ as a realization from a zero-mean Gaussian distribution with timestep-dependent variance $\Sigma_t$, yielding the following loss:

$$\mathcal{L}_{\text{physics}} = \mathcal{R}(\Delta\phi, t) = \lambda \cdot \left[ -\log p\left(\Delta\phi \mid 0, \Sigma_t\right)\right], \tag{9}$$

where $\lambda$ is a scalar coefficient that controls the contribution of the physics term. In practice, $\Sigma_t$ is obtained directly from the diffusion model's posterior variance schedule at timestep $t$, which naturally reflects the model's uncertainty over the denoising trajectory. As the reverse process proceeds and the posterior variance decreases, the likelihood sharpens, increasingly penalizing deviations from physical constraints. This construction induces an implicit time-dependent weighting. Early in the process, when uncertainty is high, the model prioritizes recovering the coarse structure of the data and allows flexibility in the residual. Later, as predictions become more confident, the loss enforces stronger adherence to physical consistency.

This formulation is particularly important for systems that combine well-understood components with poorly constrained processes. Our Eq. 8 explicitly integrates the known physical dynamics captured in Eq. 6, while treating the unknown or uncertain components as noise.

The training loss of our algorithm is then a combination of the data fidelity and physics residual term represented as follows:

$$\mathcal{L}_{\text{total}} = \mathcal{L}_{\text{data}} + \mathcal{L}_{\text{physics}}, \tag{10}$$

At inference time, we aim to reconstruct the clean state $\hat{\mathbf{x}}_0$ from a noisy initial sample $\mathbf{x}_T$ using the standard reverse diffusion process. Notably, while our model incorporates partial physical knowledge during training through the physics projection operator $\phi$, this operator is not applied during inference. This is because external physical inputs are typically unavailable for frames or timesteps that do not exist yet (*e.g.*, in prediction or infilling tasks). We generate the unknown frames by sampling from the learned reverse process while keeping the known frames fixed, following a standard clamping strategy used in conditional diffusion models [9].

## 4 Experiments

### 4.1 Data Preparation and Training

We train our PPIDM on the Chl concentration and velocity field data from the Biogeochemical Southern Ocean State Estimate dataset [21]. The dataset spans 2008 to 2012, with a spatial resolution of 1/6° and a temporal resolution of 3 days. The data consists of time, latitude, longitude and other attributes, and can be visualized as temporal continuous images in the Southern Ocean. Given the approximately log-normal distribution of Chl measurements [12], we apply a logarithmic transform during preprocessing. We gather the Chl data along with the corresponding velocity field data on the horizontal directions $u$ and $v$ of the ocean surface at each timestep. We segment each image into $64 \times 64$ patches with a sliding window, discarding any tiles containing landmass. This cropping expands the number of training samples while maintaining the essential oceanic regions. During sampling, the complete image at a given timestep can be reconstructed by independently sampling each image patch and assembling them into a full frame.

Additionally, we organize the training data by slicing across the time dimension, allowing the model to learn temporal dependencies. Specifically, for each $64 \times 64$ region, we build training sequences $\{(\mathbf{x}_n, \mathbf{x}_{n+1}, ...\mathbf{x}_{n+T-1}), (\mathbf{x}_{n+1}, \mathbf{x}_{n+2}, ..., \mathbf{x}_{n+T}), ...\}$, where $\mathbf{x}_n$ represents the Chl state at time index $n$, and $T$ represent the length of the window for training. Sliding windows that extend beyond the available temporal boundaries are discarded. The test set is designed to be the first $T$ timesteps of each spatial region, with a $2T$ timestep buffer following these test frames excluded from training to ensure independence of spatial patterns between the sequences used in training and testing. Notably, inference can be performed on a sequence of any lengths. For consistency, we set $T = 20$ for training and inference in all experiments. We train all models on a single NVIDIA RTX 4090 GPU. We set the number of data loading workers to 16, the batch size to 64, and the learning rate to $1 \times 10^{-4}$.

### 4.2 Experiment Setup

We consider two representative scenarios to evaluate the model's ability to integrate partial physical knowledge, following section 3.1. These scenarios reflect common situations in many research areas, including oceanography, where only certain aspects of the underlying dynamics are known.

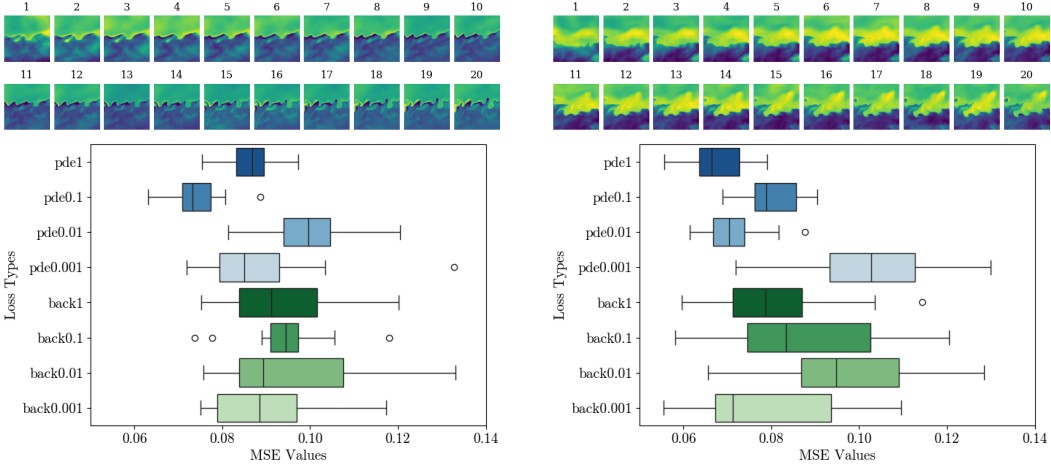

(a) Region A reconstruction MSE        (b) Region B reconstruction MSE

Figure 2: The reconstruction MSE with different weights of physics operator for different regions. Loss types have format <op><λ>, where op ∈ {pde, back} and λ ∈ {0.001, 0.01, 0.1, 1} (*pde*: ADR PDE physics operator; *back*: backtrack physics operator). Given frames 1 and 20, reconstruct frames 2-19, and repeatedly sample for 20 times to form the box plot. Different regions favor different operator-weight pairs, reflecting inter-region variability in physics dominance.

**Case 1: Partially Known ADR Parameters.** Chl dynamics at the ocean surface is modeled using the ADR PDE:

$$\frac{\partial \mathbf{x}(t)}{\partial t} + \nabla \cdot \big(\mathbf{u}(t)\,\mathbf{x}(t)\big) = \nabla \cdot \big(D(t)\,\nabla\mathbf{x}(t)\big) + R(t), \tag{11}$$

where the advection term is well specified, but the diffusion $D(\mathbf{x}, t)$ and reaction $R(\cdot)$ terms remain poorly constrained. We train our model with a constraint derived from a physics operator $\phi$ consisting of the local change rate and advection terms (*i.e.*, material derivative). This operator is applied to the $\hat{\mathbf{x}}_0$ as follows:

$$\phi\big(\hat{\mathbf{x}}_0\big) = \frac{\partial \hat{\mathbf{x}}_0(t)}{\partial t} + \nabla \cdot \big(\mathbf{u}(t)\,\hat{\mathbf{x}}_0(t)\big). \tag{12}$$

An analogous transformation $\phi(\mathbf{x}_0)$ is computed for the ground-truth field $\mathbf{x}_0$. Minimizing the discrepancy between $\phi(\mathbf{x}_0)$ and $\phi(\hat{\mathbf{x}}_0)$ encourages the model to learn predictions consistent with the known advection portion of ADR PDE, leaving diffusion and reaction terms to be inferred from data. We denote this as the *pde* physics operator.

**Case 2: Fully Known Particle-Tracking Law.** We next consider a situation where the advection of Chl parcels itself is precisely known. We can track the positions of existing parcels backward in time based on measured horizontal velocities, but parcel emergence and disappearance due to biological processes remain untractable. Specifically, for each point $(x, y)$, velocities $\mathbf{u} = (u, v)$, and small increments $\Delta t, \Delta x, \Delta y$, the law for backtracking particle positions is:

$$\begin{aligned} x_{\text{back}} &= x - u_{\text{data}}(x, y, t)\,\Delta t\,\Delta x, \\ y_{\text{back}} &= y - v_{\text{data}}(x, y, t)\,\Delta t\,\Delta y, \end{aligned} \tag{13}$$

where $x_{\text{back}}, y_{\text{back}}$ denotes the backtracked position of parcels at time $t - \Delta t$. To formalize this known transport mechanism, we define a physics operator $\phi$ that maps any field $\mathbf{x}(x, y, t)$ to its backtracked value:

$$\phi\big(\mathbf{x}(x, y, t)\big) = \mathbf{x}(x_{\text{back}}, y_{\text{back}}, t - \Delta t). \tag{14}$$

This operator is applied to both predicted and ground-truth fields. By encouraging alignment between $\phi(\hat{\mathbf{x}}_0)$ and $\phi(\mathbf{x}_0)$, the model is guided to produce predictions that are physically consistent with known particle dynamics, while still allowing flexibility to capture unresolved biological influences. We denote this as the *backtrack* physics operator.

Table 1: Overall evaluation results on test set.

| Model | Infilling | | Prediction | |
|---|---|---|---|---|
| | **RMSE** | **MAE** | **RMSE** | **MAE** |
| PIDM | 0.538 | 0.493 | 0.491 | 0.446 |
| CoCoGen | 0.460 | 0.371 | 0.309 | 0.250 |
| vanilla | 0.410 | 0.330 | 0.306 | 0.245 |
| DPS | 0.393 | 0.311 | 0.298 | 0.238 |
| **PPIDM (ours)** | **0.270** | **0.208** | **0.268** | **0.202** |

## 4.3 Results

To evaluate the performance of PPIDM, we establish four baseline comparisons of different physics integration paradigms: (i) a vanilla diffusion model trained only with data fidelity loss, (ii) a physics-informed diffusion model (PIDM) [1] that incorrectly enforces the advection term as the complete PDE to the training loss, (iii) a model following diffusion posterior sampling (DPS) [3] that injects advection information for posterior refinement only during sampling, and (iv) a model following the CoCoGen [11] framework which also uses a vanilla diffusion model and injects physics only during the last sampling steps but assumes the advection itself fully describes the system dynamics. This design of baselines isolates the effects of constraint timing of training versus sampling and physical completeness handling. We evaluate model performance using standard numerical metrics including Mean Squared Error (MSE), Root Mean Squared Error (RMSE), and Mean Absolute Error (MAE), which directly quantify deviations from the ground truth. Metrics designed to assess perceptual quality are not suitable for our task.

**Physics Operators and Weights**    To demonstrate the different effects of the physics operators with varying weights, we perform a long-range infilling task across multiple spatial regions in the test set. Two representative regions are shown in Figure 2. Notably, the optimal weight and the best choice of operator varies between regions, which reflects underlying differences in local dynamics. Region B achieves more accurate and stable reconstructions under strong advection constraints, which suggests that advection strongly dominates the dynamics in this region. In contrast, region A shows better reconstruction under moderate physics guidance, likely due to more complex or biologically modulated dynamics such as phytoplankton activation or unresolved biogeochemical processes. Therefore, to achieve a more accurate reconstruction of a given region, region-specific calibration is needed. To ensure consistency, we use the pde operator with a fixed weight of 0.1 in all reported PPIDM results.

**Comparisons of Model Performance**    To provide an overview of model performance, we report the mean RMSE and MSE across the entire test set, excluding standard deviation due to spatially heterogeneous dynamics (Table 1). To illustrate the stability of model generation, we select one representative region to sample for 20 times and report the mean and standard deviation (Tables 2 3). We focus on two core tasks: long-range spatiotemporal infilling and future-frame prediction.

To demonstrate the consequences of treating incomplete physics as fully known, the PIDM-derived baseline which minimizes the PDE residual to zero during training results in the highest reconstruction errors. By forcing the model to satisfy an oversimplified physics constraint during training, the method introduces conflicting gradients. The data fidelity term pulls solutions toward the true manifold $\mathcal{M}_{\text{data}}$, while the physics loss restricts them to an incorrect subspace $\mathcal{M}_{\text{phy}} = \{\mathbf{x} : \phi(\mathbf{x}) = 0\}$. This conflict corrupts the learned distribution, producing solutions that neither align with observations nor respect latent dynamics. Similarly, the CoCoGen-based method [11] treats partial physics as complete during inference, injecting advection constraints in the last $30\%$ of the denoising steps. Although the early generation process is unaffected, its final reconstruction with false physics assumptions misaligns with the true dynamics. This results in globally plausible layouts but locally inconsistent details.

The purely data-driven baseline achieves moderate performance. However, without physics constraints, the model generates results that only align with the learned data distribution from the limited training samples. While the model occasionally produces plausible solutions, its high variance reflects unreliable adherence to physical laws.

Table 2: Infilling performance of baseline models and our model when given only the 1st and 20th frame as input to reconstruct the intermediate 18 frames (frames 2–19). Due to space constraints, only partial frames are shown. PPIDM achieves smooth transitions that aligns best with the ground truth (GT) frames.

| GT | Frame | Models | | | | |
|---|---|---|---|---|---|---|
| | | PIDM [1] | CoCoGen [11] | vanilla | DPS [3] | **PPIDM (ours)** |
|  | 2 |  |  |  |  |  |
|  | 8 |  |  |  |  |  |
|  | 13 |  |  |  |  |  |
|  | 19 |  |  |  |  |  |
| **RMSE** | | 0.654 (0.0005) | 0.461 (0.173) | 0.460 (0.165) | 0.413 (0.138) | **0.270** (0.011) |
| **MAE** | | 0.608 (0.0006) | 0.374 (0.146) | 0.373 (0.143) | 0.333 (0.117) | **0.208** (0.016) |

Table 3: Prediction performance of baseline models and our model when given the first 10 frames to predict the next 10 future frames (frames 11–20). PPIDM achieves the best result in preserving the dynamics.

| GT | Frame | Models | | | | |
|---|---|---|---|---|---|---|
| | | PIDM [1] | CoCoGen [11] | vanilla | DPS [3] | **PPIDM (ours)** |
|  | 11 |  |  |  |  |  |
|  | 14 |  |  |  |  |  |
|  | 17 |  |  |  |  |  |
|  | 20 |  |  |  |  |  |
| **RMSE** | | 0.615 (0.0002) | 0.360 (0.051) | 0.348 (0.022) | 0.344 (0.041) | **0.266** (0.027) |
| **MAE** | | 0.571 (0.0002) | 0.308 (0.047) | 0.295 (0.015) | 0.292 (0.039) | **0.209** (0.022) |

DPS-based approaches [3] incorporate partial physics during the sampling step. Our finding confirms that integrating physics in the early diffusion steps yields poor reconstruction results, because applying deterministic PDE constraints to noisy latents produces destabilizing gradient signals. As a result, we follow the setup of CoCoGen [11] and restrict the integration of physics to the last 300 denoising steps. The results improve slightly compared to the vanilla diffusion model. However, we still observe suboptimal performance, as the denoising trajectory is already misaligned with the physics manifold by the time guidance begins, and structural errors have accumulated that a post-hoc operator cannot correct. In contrast, our PPIDM have significant improvements compared to the baseline models. It achieves the lowest average MSE by guiding the reconstructions into physical solution manifolds. Although the standard deviation is moderately higher than that observed when enforcing a complete PDE, this is reasonable because using only partial physics leaves room for unknown processes and does not strictly constrain the solution.

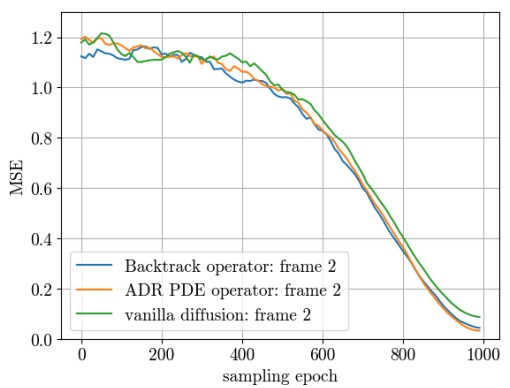 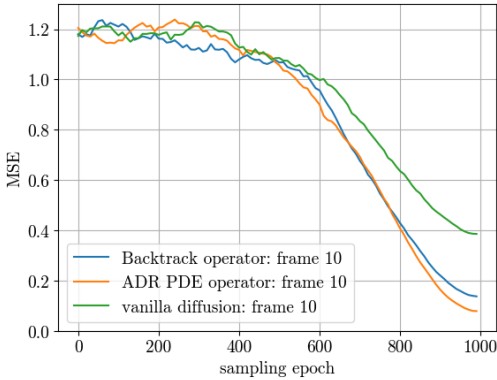

(a) Sampling step MSE for frame near the known frame(s)

(b) Sampling step MSE for frame far from the known frame(s)

Figure 3: Reconstructing frames far from the known frames benefit more from the injected partial physics knowledge (Task: given only the 1st and 20th frame as input to infill the intermediate 18 frames).

Finally, we demonstrate another key advantage of PPIDM: its robustness in reconstructing frames that are temporally distant from known reference frames, which is a setting common in long-sequence infilling and prediction tasks. In such cases, reconstruction quality typically degrades due to the limited information propagated from the observed frames. As shown in Figure 3, the difference in performance becomes more pronounced with temporal distance, which suggests that partial physics guidance becomes increasingly beneficial when generative uncertainty is high.

# 5  Conclusion and Future Work

In this paper, we present PPIDM, a framework that extends physics-informed machine learning by integrating partially known physical constraints into the training of diffusion models. Our preliminary experiments demonstrate that PPIDM outperforms both vanilla diffusion models and existing physics-informed baselines that incorrectly assume complete physical knowledge. Given the prevalence of uncertain physical knowledge across various fields, PPIDM offers a generalizable approach for incorporating such knowledge into diffusion models.

Currently, PPIDM is trained on complete time series datasets without observation noise or missing values. The method is sensitive to input data quality, so we recommend that practitioners assess data fidelity before deployment and skip missing frames when loading data rather than performing naive temporal interpolation. Future work can focus on extending the framework to accommodate systems with uncertain or incomplete observations, such as large-scale satellite-based remote sensing data of the global ocean, which often contain substantial spatiotemporal gaps. In addition, future work may explore the design of more sophisticated operators for more nuanced enforcement of physics constraints, particularly in multi-physics settings involving coupled PDEs or interacting physical subsystems.

# 6 Acknowledgments

We acknowledge the research support from Duke Kunshan University. We thank the authors of the Biogeochemical Southern Ocean State Estimate dataset [21] for providing the data.

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
