# OpenReview forum: "Partial Physics Informed Diffusion Model for Ocean Chlorophyll Concentration Reconstruction"
_NeurIPS.cc/2025/Conference — NeurIPS 2025 poster_

### Official Review · Reviewer_vCAo · 2025-06-03

**Clarity:** 3
**Significance:** 2
**Originality:** 2
**Rating:** 3
**Confidence:** 4

**Summary:**

The authors note that the physics only defines some of the behavior in their equations, and thus dx/dt = Physics(x) + Biology(x). From this they devise the partial physics-informed diffusion model, which regularizes that dx/dt is approximately Physics(x), but does not try to over constrain towards this assumption because it's know the physics is an incomplete picture. They test against vanilla machine learning models and models that constrain directly on the assumption that dx/dt = Physics(x) and show that this partial assumption outperforms both cases.

**Questions:**

1. What is the performance of this method against other partial physics methods, in particular physics-informed neural networks, universal differential equations, and physics-informed neural operators?
2. If non-Gaussian noise is applied to the data, how does that effect the performance of the method?
3. The method seems heavily reliant on hyperparameters of how much weight to put on the physics. Can some graphs be included to demonstrate that this is not something that requires much tuning, i.e. does this only work in a specific range of physics weights of does pretty much any value do well?

**Ethical Concerns:**

["NO or VERY MINOR ethics concerns only"]

**Final Justification:**

There are no benchmarks to actually determine how good the method is, so it is not clear how it performs against the state of the art.

**Limitations:**

Yes

**Quality:**

2

**Strengths And Weaknesses:**

The idea makes sense, the method does well in the benchmarks, but the testing baselines are very weak. There are many other models that can include partial physics, for example physics-informed neural networks and universal differential equations are the two most popular approaches, yet neither of those are benchmarked with the baseline.

Additionally, much of the method relies on an assumption of Gaussian noise in the derivation, but biological noise is not likely to be Gaussian. For smaller particle numbers it's generally noise to follow a Poisson process, which is only Gaussian-like in the limit as the number of particles approaches infinity. However, no testing is done to see whether non-Gaussian noise in the data causes issues with the chosen method.

---

> ### Author Rebuttal · Authors · 2025-07-30
>
> We sincerely thank reviewer vCAo for the insightful questions. We will include clarifications and experiments in the final version accordingly.
>
> ***1. Performance against partial physics baselines***
>
> PINN, PINO, and UDE are valuable physics-informed approaches. Respectfully, however, PINN, PINO and UDE have fundamental differences to our PPIDM in objective and capabilities, so we believe direct numerical comparisons are not suitable in our case:
>
> - PINN is a PDE solver that embed the *exact* PDE residual into a single neural network to solve or infer one deterministic field. In chlorophyll dynamics, they require a fully known ADR PDE including diffusion and reaction terms (biology) which are inaccessible.
> - PINO learns the solution operator of parametrized PDEs by combining sparse solution data with exact physics residuals at collocation points. They assume complete knowledge of the governing operator and cannot enforce unknown physics terms.
> - UDE augment known dynamics with a neural residual and integrate forward from an initial condition. During inference, they still require the known physics of the state to be predicted, which is inaccessible in real world. They produce only one trajectory per initial state and thus cannot support conditioning on additional states.
>
> By contrast, PPIDM is a novel *generative diffusion framework* that:
> - Requires only partial physics and no explicit laws or forms for unknown dynamics.
> - Directly samples an ensemble of plausible chlorophyll fields $x_0$, and can provide multiple physics plausible samples.
> - Enables more flexible conditioning on additional states, and can perform any state prediction and infilling.
>
> ***2. Gaussian vs Poisson noise***
>
> The reviewer has made an important point. In our work, we took a log on the chlorophyll data before normalization and training, as biology in the ocean follows a *log normal* distribution as supported by [1]. We will provide clarifications and make this transformation more explicit in the final version. This "Gaussianizes" the data and aligns well with the diffusion model’s assumption of Gaussian perturbations in the forward and reverse processes.
>
> For experimenting with Poisson noise, to avoid disrupting diffusion model's Gaussian noising assumptions, we retain the standard gaussian schedule for learning data distribution and modify the physics loss term to a Poisson negative log likelihood summed over all spatial positions $i$:
>
> $\mathcal{L}_{\mathrm{Pois}}(r,k) = \sum{i}\Bigl[r_i - k_i\log r_i + k_i\log k_i - k_i\Bigr]$, where
>
> $r = \phi(\hat{x}_0)$ is the model’s predicted rate, and $k = \phi(x_0)$ is the observed count at each diffusion step. We implemented this with the same weight for each operator and observed an average of 42.3% increase in RMSE and 45.7% increase in MAE. We believe this degradation arises because the diffusion backbone remains optimized for Gaussian noise dynamics, so imposing a Poisson‐based loss partially misaligns the learned perturbation structure.
>
> ***3. Choice of $\lambda$***
>
> Following the rebuttal rules, we apologize for not being able to show graphs and will use tables instead. Due to the difficulty to quantify physics dominance, we split our study into two broad region types, coastal and open ocean, where coastal regions are more under the affect of biological processes as supported by literature [2]. We conduct an expanded analysis across 20 diverse regions (10 coastal, 10 open ocean).
>
> To show the format of our recorded data, we first use examples of randomly sampled 2 coastal regions and 2 open ocean regions. For each region and each operator $\phi_{\rm back}, \phi_{\rm PDE}$ we trained under
> $\mathcal L = \mathcal L_{\rm data} + \lambda\\mathcal L_{\rm phys} \quad\text{with}\quad\lambda \in \{0.001\,0.01\,0.1\,1\}$.
>
> Task: Long-range infilling of frames 2–19 given frames 0 and 20; 15 independent stochastic runs per region.
>
> In the table, rows are named by the operator name and weight (for example, back0.001 means using the backtrack operator with weight 0.001), each cell is recorded as mean(std):
> | Method  |    Open Ocean Region 1 RMSE   | Open Ocean Region 1 MAE   | Open Ocean Region 2 RMSE     | Open Ocean Region 2 MAE      | Coastal Region 1 RMSE     | Coastal Region 1 MAE      |  Coastal Region 2 RMSE   |  Coastal Region 2 MAE   |
> |--|--|--|--|--|--|--|--|--|
> | back0.001| $0.303 (0.023)$ | $0.241 (0.025)$ | $0.352 (0.037)$ | $0.282 (0.023)$ | $\mathbf{0.311} (0.025)$ | $\mathbf{0.234} (0.031)$ | $\mathbf{0.257} (0.029)$ | $\mathbf{0.206} (0.026)$ |
> | back0.01 | $0.316 (0.023)$ | $0.250 (0.018)$  | $0.349 (0.039)$ | $0.280 (0.033)$ | $0.317 (0.023)$ | $0.241 (0.025)$ | $0.277 (0.025)$ | $0.224 (0.023)$ |
> | back0.1| $0.315 (0.026)$ | $0.247 (0.021)$  | $0.338 (0.027)$ | $ 0.267 (0.029)$ | $0.313 (0.016)$ | $0.235 (0.018)$ | $0.264 (0.029)$ | $0.215 (0.023)$ |
> | back1| $\mathbf{0.297} (0.019)$ | $\mathbf{0.235} (0.018)$ | $ \mathbf{0.329} (0.025)$ | $ \mathbf{0.264} (0.021)$ | $0.333 (0.026)$ | $0.251 (0.022)$ | $0.258 (0.037)$ | $0.208 (0.033)$ |
> | pde0.001| $0.320 (0.008)$ | $0.251 (0.006)$  | $0.327 (0.034)$ | $0.250 (0.018)$ | $0.331 (0.013)$ | $0.248 (0.014)$ | $\mathbf{0.205} (0.005)$ | $\mathbf{0.157} (0.005)$ |
> | pde0.01| $0.322 (0.012)$ | $0.258 (0.011)$  | $0.313 (0.020)$ | $0.248 (0.018)$ | $\mathbf{0.295} (0.010)$ | $\mathbf{0.211} (0.007)$ | $0.224 (0.013)$ | $0.171 (0.007)$ |
> | pde0.1| $0.306 (0.015)$ | $0.240 (0.012)$  | $\mathbf{0.292} (0.020)$ | $\mathbf{0.230} (0.015)$ | $0.328 (0.017)$ | $0.244 (0.017)$ | $0.242 (0.009)$ | $0.195 (0.008)$ |
> | pde1 | $\mathbf{0.301} (0.018)$ | $\mathbf{0.238} (0.016)$   | $0.317 (0.024)$ | $0.249 (0.022)$ | $0.316 (0.014)$ | $0.236 (0.018)$ | $0.242 (0.038)$ | $0.186 (0.031)$ |
> | $\Delta_{\text{Back}}$ (\%) | $6.0$ | $6.0$ | $6.5$ | $6.4$ | $6.6$ | $6.8$ | $7.2$ | $8.0$ |
> | $\Delta_{\text{PDE}}$ (\%)  | $6.5$ | $7.8$ | $10.7$ | $8.0$ | $10.9$ | $14.9$ | $15.3$ | $19.5$ |
> | vanilla diffusion | $0.441 (0.087)$ | $0.364 (0.084)$ | $0.463 (0.072)$ | $0.378 (0.076)$ | $0.413 (0.110)$ | $0.334 (0.109)$ | $0.402 (0.125)$ | $0.338 (0.114)$ |
> | $\Delta_{\text{improv}}$ (\%)  | $27.0$ | $29.1$ | $24.0$ | $25.4$ | $19.4$ | $24.9$ | $31.1$ | $33.7$ |
>
> We then report the best $\lambda$ values and the mean of deltas across 20 regions:
>
> || Coastal Region | Open Ocean Region |
> |---|---|---|
> |$\lambda$| $(0.001,0.01)$                  | $(0.1,1)$  |
>
> | $\overline{\Delta_{\text{Back}}}$ (\%) | $\overline{\Delta_{\text{PDE}}}$ (\%) |
> |---|---|
> | $7.8$                      | $14.7$  |
>
> Due to the varying balance between physical advection and biological processes across regions, we can empirically conclude that
>
> 1. Low physics-dominated areas (coastal regions) achieve their best reconstructions with small physics-loss weights, since they allow the diffusion model more freedom to capture biology.
> 2. High physics-dominated areas (open ocean) perform best with larger weights, which more strongly enforce advection consistency.
>
> In all cases, our PPIDM outperforms vanilla diffusion in both accuracy and stability by a large margin even with a small weight of the physics operator. The performance of various $\lambda$ differs within modest ranges, so domain scientists can confidently use them as starting points and fine-tune locally if needed.
>
> We thank the reviewer again for the important questions.
>
> [1] Koch, A. L. 1966. The Logarithm in Biology. Journal of Theoretical Biology, 12, 276–290.
>
> [2] Mann et al., 2006. Dynamics of Marine Ecosystems: Biological - Physical Interactions in the Oceans.

---

> > ### Comment · Reviewer_vCAo · 2025-08-03
> >
> > Some of the claims in the rebuttal are trivially untrue. For example, "PINN is a PDE solver that embed the exact PDE residual into a single neural network to solve or infer one deterministic field. In chlorophyll dynamics, they require a fully known ADR PDE including diffusion and reaction terms (biology) which are inaccessible." One of the main use cases for PINNs is to solve inverse problems, a quick search gives a few thousand results like https://www.tandfonline.com/doi/full/10.1080/17499518.2021.1971251, https://www.sciencedirect.com/science/article/pii/S004578252400882X, https://arxiv.org/abs/2407.10836, https://arxiv.org/html/2407.10654v1, etc. Solving the inverse problem by definition means learning the parameters or potentially functions of the problem, with one of the examples being a reaction-diffusion problem (of which there's about 50 publications on doing). They are even used in cases where functions are unknown as well.
> >
> > "PINO learns the solution operator of parametrized PDEs by combining sparse solution data with exact physics residuals at collocation points. They assume complete knowledge of the governing operator and cannot enforce unknown physics terms." also simply not true given neural operators are used in inverse problems as well https://arxiv.org/abs/2301.11167, https://arxiv.org/abs/2402.11722, https://proceedings.neurips.cc/paper_files/paper/2024/file/39f6d5c2e310a5a629dcfc4d517aa0d1-Paper-Conference.pdf, etc.
> >
> > "UDE augment known dynamics with a neural residual and integrate forward from an initial condition. During inference, they still require the known physics of the state to be predicted, which is inaccessible in real world. They produce only one trajectory per initial state and thus cannot support conditioning on additional states." once again, not true. In https://arxiv.org/abs/2001.04385 example 2 is learning to find the difference between PDE operators via convolutional kernels, and https://www.youtube.com/watch?v=tit1lQTSSXI has cases for example where graphs are used as prior knowledge when there are no known equations.
> >
> > So,
> >
> > " By contrast, PPIDM is a novel generative diffusion framework that: Requires only partial physics and no explicit laws or forms for unknown dynamics., Directly samples an ensemble of plausible chlorophyll fields $x_0$, and can provide multiple physics plausible samples., Enables more flexible conditioning on additional states, and can perform any state prediction and infilling."
> >
> > All apply to the methods discussed, and the authors didn't test against baselines there so it's hard to know how it performs against the other methods of the field.
> >
> > And the shown results show that lambda fine-tuning does matter, there isn't a default that is universally used.
> >
> > I don't think there's anything technically wrong with the method, but it's just not clear there's any benefit to it from what came before.

---

> ### Author Response · Authors · 2025-08-04
>
> We thank reviewer vCAo for the detailed comment.
>
> We agree with the reviewer that PINN, PINO and UDE can handle partial physics in inverse problem settings. Once PINN, PINO and UDE learn the differential equations, they can integrate forward to perform prediction. However, we want to point out that PPIDM and these methods defer in fundamental goals and makes comparisons in single scenarios unfair.
>
> Unlike PINN, PINO and UDE based methods, PPIDM is designed to **generate missing spatiotemporal samples in tasks such as forward predictions, backward extrapolations or interpolation between arbitrary frames** so that they are as physically plausible as possible. To achieve this, PPIDM can condition on **any** available observations at inference time. As an intuitive example, given chlorophyll distribution at timestep 1,6,7,8,9, the task is to infer the distribution at timestep 2,3,4,5,10 in one pass. While PINN, PINO and UDE based methods can only accept the initial state (timestep 1) and integrate forward to perform prediction, PPIDM can condition on the full subset of known states. Given these discrepancies, comparing PPIDM to the mentioned methods would be unfair, as the comparison can only be designed either to underutilize PPIDM's flexibility (give only distribution at timestep 1 as condition), or give extra advantage to PPIDM (condition it on distributions of all known timesteps). PPIDM is flexible enough so that once trained, it can infer any number of unknown frames conditioned on any subset of known frames, without retraining. Thus, PPIDM's novelty compared to the mentioned methods is in real world deployment of reconstructing complex spatiotemporal datasets under partial physics and irregular observations.
>
> Also, we agree that $\lambda$ tuning is necessary for optimal performance, which is universal for machine learning methods in different tasks. Importantly, we showed that the performance of models trained with different $\lambda$ values can all outperform vanilla diffusion methods significantly, so using any lambda for scientists can all be better choices over using just vanilla diffusion model. We also demonstrated a correlation between known physics dominance state of the system and the optimal lambda, which can serve as a helpful guide for tuning.

---

### Official Review · Reviewer_vs2C · 2025-06-30

**Clarity:** 4
**Significance:** 4
**Originality:** 3
**Rating:** 5
**Confidence:** 5

**Summary:**

This paper proposes a novel Partially-Physics Informed Diffusion Model (PPIDM) for reconstructing ocean chlorophyll concentration fields when only partial physical laws are known.The model incorporates known physical processes (e.g., advection) through designed physics operators, which participate in the loss function by combining residual differences between real and predicted samples in physical projection space with data loss. Meanwhile, unknown dynamics (e.g., biological processes) are treated as noise, avoiding existing methods' over-reliance on complete physical constraints. Experiments demonstrate that PPIDM significantly outperforms baseline methods (e.g., PIDM, CoCoGen) in reconstruction accuracy and stability, particularly excelling in filling temporal gaps and predicting future values.

**Questions:**

Please refer to the weakness.

**Ethical Concerns:**

["NO or VERY MINOR ethics concerns only"]

**Final Justification:**

The author's response has effectively addressed my concerns. Although other reviewers raised some doubts, in many AI for Science problems, we only know partial physical laws. The author has provided a concise and effective solution, which is why I maintain a positive attitude toward the paper. Considering that the comments from the other reviewers are also constructive, I recommend a score of weak accept or accept for this paper.

**Limitations:**

Yes.

**Quality:**

3

**Strengths And Weaknesses:**

**Strength**
1. The integration of physical laws with uncertainties in ML algorithms is indeed understudied. The authors give a good result to solve this important problem. I really appreciate this work.
2. The approach guides the generative model through physical residuals rather than enforcing full physical constraints, which better aligns with real-world scenarios where many definitive laws remain undetermined.
3. The model demonstrates practical significance, particularly in its scientific value for analyzing Southern Ocean chlorophyll distributions.
4. The experimental design rigorously considers both physics-dominated and non-physics-dominated scenarios, with results showing substantial improvements over comparable models.

**Weakness**
1. How to design the model architecture of $\phi$? How to ensure that $\phi$ extracts and constrain the learning process under partial-known law?
2. The sensitivity study of loss balance factor $\lambda$ is missing.

---

> ### Author Rebuttal · Authors · 2025-07-29
>
> We sincerely thank reviewer vs2C for the appreciation in our work and the insightful questions. We will add the corresponding clarifications and experiments in the final version.
>
> ***1. Design of $\phi$***
>
> The physics operator $\phi$ captures accessible dynamics and is designed to reflect the evolution or constraint imposed by the known part of the physics. Depending on the available knowledge, domain scientists can design the $\phi$ of their own field by plugging in the state of interest $x_0$ into a subcomponent of a governing PDE or a partial law the system follows. Specifically,
>
> **(a) Subcomponent of a governing PDE**
>
> In many systems, the dynamics are partially described by a PDE of the form:
> $\frac{\partial x}{\partial t} = \mathcal{P}{\text{known}}(x) + \mathcal{P}{\text{unknown}}(x).$
> In this case, $\phi(x_0)$ computes the residual of $\mathcal{P}{\text{known}}$, which is $\phi(x_0) = \left| \frac{\partial x_0}{\partial t} - \mathcal{P}{\text{known}}(x_0) \right| $ to measure how much the system deviates from what would be expected if only the known physics were active.
>
> If for the predicted $\hat{x}_0$, $\phi(\hat{x}_0)$ returns a large residual, the model’s output is inconsistent with known physics. However, since the system also evolves under unknown processes $\mathcal{P}{\text{unknown}}$, the residual $\phi(\hat{x}_0)$ alone cannot be minimized directly or interpreted as absolute error. Instead, we minimize the discrepancy between $\phi(\hat{x}_0)$ and $\phi(x_0)$ across diffusion steps, which encourages the model to stay consistent with the known physics to the same extent as the real data, without over-constraining it where the dynamics are incomplete.
>
> **(b) Partial law of the system**
>
> $\phi$ can also encode domain specific laws such as mass conservation laws, fluid transport, or geometric constraints. These laws may not define the full system evolution, but they represent structural properties the state should approximately satisfy.
>
> 1. If the law is evolution-based like in our chlorophyll parcel transportation application, $\phi(x_0)$ transforms a state $x_0$ into its physically consistent counterpart at another timestep, such as the previous location of a fluid parcel if only known transport governed its motion. In this case, $\phi(\hat{x}_0)$ approximates what the state would have been under pure known physics, and we can constrain the model by minimizing its deviation from a reference (e.g., the backtracked ground truth).
>
> 2. If the law is constraint-based, $\phi(x_0)$ measures the degree of violation of the known constraint. In many real-world or partially known systems, these constraints are not strictly obeyed, and even the ground truth may show leakage, growth, or dissipation. Therefore, there is no fixed value for $\phi(\hat{x}_0)$, and can only be obtained by calculating the discrepancy from $\phi(x_0)$.
>
> Rather than enforcing strict adherence to an incomplete law, this method encourages the model to match the degree of physical consistency observed in the real system.
>
> ***2. Sensitivity study of $\lambda$***
>
> We agree that sensitivity of the model to the physics weight $\lambda$ is crucial. Due to the difficulty to quantify physics dominance, we split our study into two broad region types, coastal and open ocean, where coastal regions are more under the affect of biological processes as supported by literature [1]. We conduct an expanded analysis across 20 diverse regions (10 coastal, 10 open ocean).
>
> To show the format of our recorded data, we first use examples of randomly sampled 2 coastal regions and 2 open ocean regions. For each region and each operator $\phi_{\rm back}, \phi_{\rm PDE}$ we trained under
> $\mathcal L = \mathcal L_{\rm data} + \lambda\\mathcal L_{\rm phys} \quad\text{with}\quad\lambda \in \{0.001\,0.01\,0.1\,1\}$.
>
> Task: Long-range infilling of frames 2–19 given frames 0 and 20; 15 independent stochastic runs per region.
>
> In the table, rows are named by the operator name and weight (for example, back0.001 means using the backtrack operator with weight 0.001), each cell is recorded as mean(std). We calculate the performance variance for different weights of each operator, as well as the improvement over vanilla diffusion model for the *worst* performing operator:
> | Method  |    Open Ocean Region 1 RMSE   | Open Ocean Region 1 MAE   | Open Ocean Region 2 RMSE     | Open Ocean Region 2 MAE      | Coastal Region 1 RMSE     | Coastal Region 1 MAE      |  Coastal Region 2 RMSE   |  Coastal Region 2 MAE   |
> |--|--|--|--|--|--|--|--|--|
> | back0.001| $0.303 (0.023)$ | $0.241 (0.025)$ | $0.352 (0.037)$ | $0.282 (0.023)$ | $\mathbf{0.311} (0.025)$ | $\mathbf{0.234} (0.031)$ | $\mathbf{0.257} (0.029)$ | $\mathbf{0.206} (0.026)$ |
> | back0.01 | $0.316 (0.023)$ | $0.250 (0.018)$  | $0.349 (0.039)$ | $0.280 (0.033)$ | $0.317 (0.023)$ | $0.241 (0.025)$ | $0.277 (0.025)$ | $0.224 (0.023)$ |
> | back0.1| $0.315 (0.026)$ | $0.247 (0.021)$  | $0.338 (0.027)$ | $ 0.267 (0.029)$ | $0.313 (0.016)$ | $0.235 (0.018)$ | $0.264 (0.029)$ | $0.215 (0.023)$ |
> | back1| $\mathbf{0.297} (0.019)$ | $\mathbf{0.235} (0.018)$ | $ \mathbf{0.329} (0.025)$ | $ \mathbf{0.264} (0.021)$ | $0.333 (0.026)$ | $0.251 (0.022)$ | $0.258 (0.037)$ | $0.208 (0.033)$ |
> | pde0.001| $0.320 (0.008)$ | $0.251 (0.006)$  | $0.327 (0.034)$ | $0.250 (0.018)$ | $0.331 (0.013)$ | $0.248 (0.014)$ | $\mathbf{0.205} (0.005)$ | $\mathbf{0.157} (0.005)$ |
> | pde0.01| $0.322 (0.012)$ | $0.258 (0.011)$  | $0.313 (0.020)$ | $0.248 (0.018)$ | $\mathbf{0.295} (0.010)$ | $\mathbf{0.211} (0.007)$ | $0.224 (0.013)$ | $0.171 (0.007)$ |
> | pde0.1| $0.306 (0.015)$ | $0.240 (0.012)$  | $\mathbf{0.292} (0.020)$ | $\mathbf{0.230} (0.015)$ | $0.328 (0.017)$ | $0.244 (0.017)$ | $0.242 (0.009)$ | $0.195 (0.008)$ |
> | pde1 | $\mathbf{0.301} (0.018)$ | $\mathbf{0.238} (0.016)$   | $0.317 (0.024)$ | $0.249 (0.022)$ | $0.316 (0.014)$ | $0.236 (0.018)$ | $0.242 (0.038)$ | $0.186 (0.031)$ |
> | $\Delta_{\text{Back}}$ (\%) | $6.0$ | $6.0$ | $6.5$ | $6.4$ | $6.6$ | $6.8$ | $7.2$ | $8.0$ |
> | $\Delta_{\text{PDE}}$ (\%)  | $6.5$ | $7.8$ | $10.7$ | $8.0$ | $10.9$ | $14.9$ | $15.3$ | $19.5$ |
> | vanilla diffusion | $0.441 (0.087)$ | $0.364 (0.084)$ | $0.463 (0.072)$ | $0.378 (0.076)$ | $0.413 (0.110)$ | $0.334 (0.109)$ | $0.402 (0.125)$ | $0.338 (0.114)$ |
> | $\Delta_{\text{improv}}$ (\%)  | $27.0$ | $29.1$ | $24.0$ | $25.4$ | $19.4$ | $24.9$ | $31.1$ | $33.7$ |
>
> We then report the best $\lambda$ values and the mean of deltas across 20 regions:
>
> || Coastal Region | Open Ocean Region |
> |---|---|---|
> |$\lambda$| $(0.001,0.01)$                  | $(0.1,1)$  |
>
> | $\overline{\Delta_{\text{Back}}}$ (\%) | $\overline{\Delta_{\text{PDE}}}$ (\%) |
> |---|---|
> | $7.8$                      | $14.7$  |
>
> Due to the varying balance between physical advection and biological processes across regions, we can empirically conclude that
>
> 1. Low physics-dominated areas (coastal regions) achieve their best reconstructions with small physics-loss weights, since they allow the diffusion model more freedom to capture biology.
> 2. High physics-dominated areas (open ocean) perform best with larger weights, which more strongly enforce advection consistency.
>
> In all cases, our PPIDM outperforms vanilla diffusion in both accuracy and stability by a large margin even with a small weight of the physics operator. The performance of various $\lambda$ differs within modest ranges, so domain scientists can confidently use them as starting points and fine-tune locally if needed.
>
> We thank the reviewer again for the appreciation and hope this clarifies their questions.
>
> [1] Mann et al., 2006. Dynamics of Marine Ecosystems: Biological - Physical Interactions in the Oceans.

---

> > ### Comment · Reviewer_vs2C · 2025-08-01
> >
> > Thank you for the responses. I will continue to maintain a positive opinion.

---

> > > ### Author Response · Authors · 2025-08-04
> > >
> > > We appreciate the reviewer's acknowledgement and valuable comments.

---

### Official Review · Reviewer_hTSZ · 2025-07-02

**Clarity:** 2
**Significance:** 1
**Originality:** 2
**Rating:** 3
**Confidence:** 3

**Summary:**

This paper introduces the Partial-Physics Informed Diffusion Model (PPIDM), a novel framework designed to integrate incomplete but known physical principles into the training of diffusion models. The primary application demonstrated is the reconstruction of ocean surface chlorophyll concentration, a system governed by both well-constrained physical processes and poorly constrained biological processes. The proposed method involves defining a "physics operator" that captures only the known parts of the system's dynamics. The study present experimental results on a dataset of Southern Ocean chlorophyll concentration, showing that PPIDM outperforms several baselines, including a purely data-driven diffusion model and models that enforce an incomplete physical law.

**Questions:**

1.The choice of the physics operator and its corresponding weight seems to be a critical hyperparameter that varies between different geographical regions. Could you elaborate on the process for selecting these hyperparameters? A more detailed sensitivity analysis across a wider range of regions would strengthen the paper.
2.The "backtrack physics operator" is not clearly explained. Could you provide a more detailed mathematical derivation and a clearer explanation of how it enforces physical consistency? An illustrative example would be very helpful.
3.Your comparison with the DPS baseline involves a significant modification to the original method. Could you provide results using a more faithful implementation of DPS? A fair and rigorous comparison to existing state-of-the-art methods is essential for demonstrating the superiority of your proposed approach.

**Ethical Concerns:**

["NO or VERY MINOR ethics concerns only"]

**Limitations:**

1. The authors have not adequately addressed the limitations and potential negative societal impacts of their work. The "Conclusion and Future Work" section briefly mentions that the model is trained on complete data, but it does not explore the potential failure modes of the model when faced with noisy or incomplete real-world data.
2. The authors should include a dedicated "Limitations" section and discuss the sensitivity of their model to the quality of the input data. For example, how does the model perform if the velocity field data (used in the physics operator) is noisy or has low resolution?
3. The authors should also discuss the potential for their model to produce physically plausible but incorrect reconstructions, especially in regions where the unmodeled biological processes are dominant. This could have negative consequences if the model's outputs are used for making decisions about ecosystem management or climate policy.

**Quality:**

2

**Strengths And Weaknesses:**

Strength:
- The core idea of using a "physics residual difference" rather than forcing a physics residual to zero is a sensible and well-motivated approach for problems with incomplete physical models.
- The overall structure of the paper is logical, and the introduction provides a good motivation for the problem being addressed.
- The application of these ideas to diffusion models for spatiotemporal reconstruction is a novel and interesting research direction.

Weakness:
- The experimental evaluation is not robust enough to fully support the paper's claims. The results are presented for only two "representative" regions, and the authors themselves note that the optimal choice of physics operator and its weighting varies between these regions. This suggests that the method may require significant, region-specific tuning, which undermines its generalizability. A more thorough evaluation across a much larger and more diverse set of regions is needed to demonstrate the robustness of the proposed approach.
- The figures are not well-explained and are difficult to interpret. Figure 2, for example, presents a complex set of results with different operators and weights, but the caption and main text provide insufficient guidance for the reader to fully understand its implications. The box plots are also not clearly explained.
- While the specific application to diffusion models is novel, the idea of incorporating partial physical knowledge into machine learning models is not new. The paper fails to adequately contextualize its contribution within the broader field of physics-informed machine learning. A more comprehensive review of related work would help to clarify the paper's unique contribution.
- The practical significance of the work is limited by the weaknesses in the experimental evaluation. Without a more convincing demonstration of the method's robustness and generalizability, its potential impact on the field remains questionable.

---

> ### Author Rebuttal · Authors · 2025-07-30
>
> We sincerely thank reviewer hTSZ for the questions and valuable suggestions. We will add more clarifications and experiments in the final version correspondingly.
>
> ***1. Visualization of representative regions and choice of hyperparameters***
>
> We agree that regional sensitivity requires deeper analysis. Due to the difficulty to quantify physics dominance, we classify the region dynamics into two types, coastal and open ocean, where coastal regions are more under the affect of biological processes as supported by literature [0]. In addition to the initial 2-region visualization, we conducted an expanded analysis across 20 diverse regions (10 coastal, 10 open ocean).
>
> We first show the format of our recorded data using examples of randomly sampled 2 coastal regions and 2 open ocean regions. For each region and each operator $\phi_{\rm back}, \phi_{\rm PDE}$ we trained under
> $\mathcal L = \mathcal L_{\rm data} + \lambda\\mathcal L_{\rm phys} \quad\text{with}\quad\lambda \in \{0.001\,0.01\,0.1\,1\}$.
>
> Task: Long-range infilling of frames 2–19 given frames 0 and 20; 15 independent stochastic runs per region.
>
> In the table, rows are named by the operator name and weight (for example, back0.001 means using the backtrack operator with weight 0.001), each cell is recorded as mean(std). We calculate the performance variance for different weights of each operator, as well as the improvement over vanilla diffusion model for the *worst* performing operator:
> | Method  |    Open Ocean Region 1 RMSE   | Open Ocean Region 1 MAE   | Open Ocean Region 2 RMSE     | Open Ocean Region 2 MAE      | Coastal Region 1 RMSE     | Coastal Region 1 MAE      |  Coastal Region 2 RMSE   |  Coastal Region 2 MAE   |
> |--|--|--|--|--|--|--|--|--|
> | back0.001| $0.303 (0.023)$ | $0.241 (0.025)$ | $0.352 (0.037)$ | $0.282 (0.023)$ | $\mathbf{0.311} (0.025)$ | $\mathbf{0.234} (0.031)$ | $\mathbf{0.257} (0.029)$ | $\mathbf{0.206} (0.026)$ |
> | back0.01 | $0.316 (0.023)$ | $0.250 (0.018)$  | $0.349 (0.039)$ | $0.280 (0.033)$ | $0.317 (0.023)$ | $0.241 (0.025)$ | $0.277 (0.025)$ | $0.224 (0.023)$ |
> | back0.1| $0.315 (0.026)$ | $0.247 (0.021)$  | $0.338 (0.027)$ | $ 0.267 (0.029)$ | $0.313 (0.016)$ | $0.235 (0.018)$ | $0.264 (0.029)$ | $0.215 (0.023)$ |
> | back1| $\mathbf{0.297} (0.019)$ | $\mathbf{0.235} (0.018)$ | $ \mathbf{0.329} (0.025)$ | $ \mathbf{0.264} (0.021)$ | $0.333 (0.026)$ | $0.251 (0.022)$ | $0.258 (0.037)$ | $0.208 (0.033)$ |
> | pde0.001| $0.320 (0.008)$ | $0.251 (0.006)$  | $0.327 (0.034)$ | $0.250 (0.018)$ | $0.331 (0.013)$ | $0.248 (0.014)$ | $\mathbf{0.205} (0.005)$ | $\mathbf{0.157} (0.005)$ |
> | pde0.01| $0.322 (0.012)$ | $0.258 (0.011)$  | $0.313 (0.020)$ | $0.248 (0.018)$ | $\mathbf{0.295} (0.010)$ | $\mathbf{0.211} (0.007)$ | $0.224 (0.013)$ | $0.171 (0.007)$ |
> | pde0.1| $0.306 (0.015)$ | $0.240 (0.012)$  | $\mathbf{0.292} (0.020)$ | $\mathbf{0.230} (0.015)$ | $0.328 (0.017)$ | $0.244 (0.017)$ | $0.242 (0.009)$ | $0.195 (0.008)$ |
> | pde1 | $\mathbf{0.301} (0.018)$ | $\mathbf{0.238} (0.016)$   | $0.317 (0.024)$ | $0.249 (0.022)$ | $0.316 (0.014)$ | $0.236 (0.018)$ | $0.242 (0.038)$ | $0.186 (0.031)$ |
> | $\Delta_{\text{Back}}$ (\%) | $6.0$ | $6.0$ | $6.5$ | $6.4$ | $6.6$ | $6.8$ | $7.2$ | $8.0$ |
> | $\Delta_{\text{PDE}}$ (\%)  | $6.5$ | $7.8$ | $10.7$ | $8.0$ | $10.9$ | $14.9$ | $15.3$ | $19.5$ |
> | vanilla diffusion | $0.441 (0.087)$ | $0.364 (0.084)$ | $0.463 (0.072)$ | $0.378 (0.076)$ | $0.413 (0.110)$ | $0.334 (0.109)$ | $0.402 (0.125)$ | $0.338 (0.114)$ |
> | $\Delta_{\text{improv}}$ (\%)  | $27.0$ | $29.1$ | $24.0$ | $25.4$ | $19.4$ | $24.9$ | $31.1$ | $33.7$ |
>
> We then report the best $\lambda$ values and the mean of deltas across 20 regions:
>
> || Coastal Region | Open Ocean Region |
> |---|---|---|
> |$\lambda$| $(0.001,0.01)$                  | $(0.1,1)$  |
>
> | $\overline{\Delta_{\text{Back}}}$ (\%) | $\overline{\Delta_{\text{PDE}}}$ (\%) |
> |---|---|
> | $7.8$                      | $14.7$  |
>
> Due to the varying balance between physical advection and biological processes across regions, we can empirically conclude that
>
> 1. Low physics-dominated areas (coastal regions) achieve their best reconstructions with small physics-loss weights, since they allow the diffusion model more freedom to capture biology.
> 2. High physics-dominated areas (open ocean) perform best with larger weights, which more strongly enforce advection consistency.
>
> In all cases, our PPIDM outperforms vanilla diffusion in both accuracy and stability by a large margin even with a small weight of the physics operator. The performance of various $\lambda$ differs within modest ranges, so domain scientists can confidently use them as starting points and fine-tune locally if needed.
>
> ***2. Figure clarity*** We will revise all captions to (a) define every marker, axis name and (b) discuss key takeaways and interpretations.
>
> ***3. Contribution in physics informed ML***
> We will extend Section 2.2 to discuss related methods that integrate partial physics, including: [1] Hybrid NN-PDE solvers correcting outputs of incomplete PDEs with NN, [2] UDEs embedding known operators while learning unknown residuals via NN, and [3] APHYNITY that uses a differentiable ODE/PDE integrator to enforce a known physics term to provide a baseline trajectory and a neural residual to explain the remaining discrepancy. However, these methods' main purpose is to *learn the PDE* under partial physics, which produce only one trajectory per initial state and thus cannot support conditioning on additional states. We will also stress our main novelty: **PPIDM is the first to (1) embed partial physics into stochastic diffusion processes, (2) support more flexible conditioning, so more robust to real world applications such as any state prediction and infilling.**
>
> ***4. Mechanism of the backtrack operator***
>
> The backtrack operator is built on the discrete continuity equation for passive tracers in a velocity field. Specifically, it computes where a given chlorophyll parcel would be at the last timestep if only ocean currents (velocity field) moved it, ignoring biological growth/decay.
>
> Derivation: Starting from the continuity equation $\frac{\partial c}{\partial t} + u \frac{\partial c}{\partial x} + v \frac{\partial c}{\partial y} = 0,$ we use a first-order approximation of the characteristic displacement induced by advection $c(t - \Delta t) \approx c(t) - \Delta t \left( u \frac{\partial c}{\partial x} + v \frac{\partial c}{\partial y} \right).$ This is equivalent to shifting chlorophyll on each point $(x, y)$ along the negative advection vector weighted by the local chlorophyll gradients: $x_{\text{back}} = x - u \Delta t \cdot \frac{\partial c}{\partial x}, \quad y_{\text{back}} = y - v \Delta t \cdot \frac{\partial c}{\partial y}$ which enforces chlorophyll flux.
>
> Example: For chlorophyll at day 10 ($x_{10}$), $\phi_{\text{back}}(x_{10})$ estimates how chlorophyll would distribute at day 9 if only currents moved it (ignoring biology). Because $\phi_{\rm back}(x_{10})\neq x_{9}$ (biology is missing), we do not compare $\phi_{\rm back}(x_{10})$ to the true $x_{9}$. Instead, at each training step we apply $\phi_{\rm back}$ to both the predicted $\hat x_{10}$ and the true $x_{10}$, and minimize their difference. This penalizes any discrepancy in how the model transports chlorophyll.
>
> ***4. DPS implementation***
>
> Our implementation of DPS strictly follows their official implementation with only modification in the measurement: 1. We replace the original imaging operator $A(\cdot)$ with our proposed physics operator $\phi(\cdot)$. 2. We set $y = \phi(x_0) $ as the measurement, in place of the original y. We train the model without physics and during sampling at each timestep we first apply the Gaussian denoise step, then compute $\nabla_x \||y - \phi(\hat{x}_0)\||^2$ and take a small gradient step, which is identical to Chung et al.
>
> ***5. Limitation***
>
> We thank the reviewer for raising these important points. In a new Limitations section, we will state explicitly that our PPIDM is trained and evaluated on fully resolved, noise‑free velocity fields. We will recommend domain scientists to assess input‐data quality before deploying our model. Failure modes includes high frequency noise and large temporal gaps. Therefore, if dealing with highly incomplete physics data, we recommend (i) pre‑processing velocity fields with simple low‑pass filters before applying $\phi$, and (ii) skipping missing frames in the data loader rather than naively interpolating them. We will add warning against over-reliance on model outputs especially in extreme biological events and recommend pairing our work with domain expert insights when used for policy making.
>
> We thank the reviewer again for the insightful suggestions.
>
> [0] Mann et al., 2006. Dynamics of Marine Ecosystems: Biological - Physical Interactions in the Oceans.
>
> [1] Tathawadekar et al., 2021. Incomplete to Complete Multiphysics Forecasting: A Hybrid Approach for Learning Unknown Phenomena. Data‑Centric Engineering, 4(31), e27.
>
> [2] Rackauckas et al., 2020. Universal Differential Equations for Scientific Machine Learning.
>
> [3] Yin et al., 2021. Augmenting Physical Models with Deep Networks for Complex Dynamics Forecasting. ICLR 2021/J. Stat. Mech., 2021(12).

---

### Official Review · Reviewer_a71N · 2025-07-03

**Clarity:** 2
**Significance:** 2
**Originality:** 3
**Rating:** 4
**Confidence:** 4

**Summary:**

The paper presents a novel framework called the Partial-Physics Informed Diffusion Model (PPIDM). The key contribution of this paper is integrating partial physical laws into the diffusion model to improve the reconstruction of chlorophyll concentration in the Southern Ocean.

**Questions:**

Can you explain exactly how you physically-informed transformation is implemented, is it calculating the residuals of the known parts of the PDE.

**Ethical Concerns:**

["NO or VERY MINOR ethics concerns only"]

**Final Justification:**

The author's response answered my doubts and I slightly improved my score

**Limitations:**

Yes, the authors discuss relevant limitations and future work.

**Paper Formatting Concerns:**

No Formatting Concerns.

**Quality:**

2

**Strengths And Weaknesses:**

**Strengths**

1. The problem examined in this article is important because there are many application scenarios, such as climate-ocean modeling, where we do not have access to accurate physical priors.

2. The article summarizes the relevant work well, and the baseline is recent and competitive.

**Weaknesses**

1. I'm having a hard time understanding what the authors mean by *physically-informed transformation*, and what *physically-informed subspace* is, and exactly how the algorithm operates, and there is no additional information on this. If I understand correctly, the final loss is still a combination of the denoising loss and the residuals of the known physics information, which seems to be similar to PIDM.

2. The authors have only experimented on the data of marine chlorophyll, and I would prefer to recommend the authors to submit the manuscript to a relevant applied professional journal where a more specialized review is available.

---

> ### Author Rebuttal · Authors · 2025-07-27
>
> We sincerely thank reviewer a71N for the comments and suggestions. We here provide more clarifications to the raised weaknesses and questions.
>
> ***1. Clarifying Physics-Informed Transformation & Relationship to PIDM***
>
> The physics-informed transformation applied to $x_0$ (whether ground-truth $x_0$ or the predicted $\hat{x}_0$ at a diffusion step) is defined as $\phi(x_0)$, where $\phi$ represents the known physics operator. As shown in Section 4.2, $\phi$ captures accessible dynamics, which can either be a subcomponent of a governing PDE or a partial law the system follows. For any state $x_0$, this operator projects it into a physics-informed subspace, that is, the set of states that are physically consistent under the known but incomplete dynamics, depicting the state *if only known physics governed the system*. For instance:
> - If only the advection term is known in the ADR PDE, $\phi(x_0)$ computes the residual of the known physics. In this case, the physics-informed subspace is defined by all states that yield zero residual under the known advection operator.
> - Fluid parcels following the continuity equation can be tracked backward along the velocity field by one timestep. $\phi(x_0)$ computes where chlorophyll would be at the last timestep if only ocean currents (velocity field) moved it, ignoring biological growth/decay. Here, the physics-informed subspace consists of states reachable by pure velocity field alone.
>
> However, since $\phi$ does *not* fully describe the system’s dynamics, minimizing the residual or enforcing strict consistency with ground-truth physics is infeasible. Instead, **PPIDM** computes $\phi(\hat{x}_0)$ and $\phi(x_0)$ at each diffusion step and minimizes their adaptive, timestep-dependent discrepancy, and integrate this with the denoising loss.
>
> This fundamentally differs from **PIDM**, which assumes *complete* physics knowledge and directly minimizes PDE residuals as a fixed loss term. PIDM’s assumption impedes learning in systems with partially unknown dynamics such as for ocean chlorophyll, where biological processes are non-negligible. **PPIDM**’s novelty lies in integrating *incomplete* physics without suppressing unmodeled dynamics, thereby enhancing generalization where PIDM struggles. Thus, while PIDM and PPIDM both combine denoising and physics losses, their mechanisms and assumptions diverge significantly.
>
> ***2. Scope and Application Focus***
>
> We thank the reviewer for the suggestion, however, we submit our paper to this venue for a wider exposure to help the broader field of researchers in the ML for science community due to:
> - Ocean chlorophyll dynamics are an ideal example for partial physics, as it is dominated by well-understood ocean currents (advection) but also significantly influenced by poorly quantified biological processes.
> - Focusing on one complex, real-world problem allows us to rigorously develop and evaluate PPIDM's core mechanics.
> - PPIDM provides a easily generalizable framework. Domain scientists can plug in their specific $\phi$ (representing their partial knowledge) into the diffusion process using our approach.
>
> We thank the reviewer again for the constructive feedback and hope the clarifications can help resolve issues. We will revise the final version accordingly to provide clearer explanations.

---

> > ### Comment · Reviewer_a71N · 2025-08-06
> >
> > Thank you for the author's response. I seem to have a clearer understanding of the proposed method from your response. I still have some doubts. As far as I know, due to discretization errors, even with complete physical knowledge, $\phi (x_0) $is still not zero. Therefore, $\phi (x_0) $may be composed of two parts rather than just unknown dynamics. How does the author view this.

---

> > > ### Author Response · Authors · 2025-08-07
> > >
> > > We thank the reviewer for the question and we agree that discretization error exists. Based on insights from PIDM, in our framework we model the physics loss as a zero mean Gaussian with covariance $\Sigma_t$. This is not a hard constraint that forces the physics loss to zero, but instead can tolerate small discretization errors while penalizing larger deviations. We will make this tolerance to discretization error explicit in the final version.

---

### Note · Authors · 2025-08-12

We thank the reviewers, ACs, and SACs for your efforts in our work.

We would like to highlight our core contributions:
1) PPIDM integrates **incomplete** physics knowledge into the diffusion process via a novel physics residual difference.
2) Can condition on **any** subset of observations during inference, therefore more flexible for real-world spatiotemporal reconstruction.
3) Provides parameter tuning guide to account for different dominance states of known physics.

In the final version, we will incorporate the reviewers’ constructive feedback by:
1) **More clarifications on terminology and mechanism**: We will expand Section 3.2 to explicitly define key terms such as physics-informed transformation and physics-informed subspace. We will clarify the design and mechanism of the physics operator in each known-physics case, including how it extracts and constrains learning under partial knowledge and flexibility to discretization error.
2) **More related works of physics informed ML**: We will expand Section 2.2 with analysis of partial-physics informed ML baselines and clarify their differences in objectives and inference flexibility compared to our PPIDM.
3) **Incorporating sensitivity analysis on $\lambda$**: We will add results to show correlations between physics dominance and optimal $\lambda$. To demonstrate robustness, we will discuss that all $\lambda$ settings outperform vanilla diffusion significantly even without fine-tuning.
4) **Improving visualization readability**: We will revise captions for figures 1-3 and tables 1-3 to fully define markers, axes, and include key takeaways to ensure complex plots and tables are interpretable.
5) **Adding limitations and social impact**: We will include a limitations section noting sensitivity to input data quality and caution against over-reliance especially in biology-dominated extremes without domain-expert validation.

Thank you again for the valuable insights.

---

### Decision · Program_Chairs · 2025-09-17

**Decision:**

Accept (poster)

**Comment:**

This paper presents the Partial-Physics Informed Diffusion Model (PPIDM), which integrates incomplete but known physical principles into diffusion models. The application to ocean surface chlorophyll reconstruction is both timely and relevant, as many scientific domains are governed by partially understood dynamics. The rebuttal strengthened the work with additional experiments, clarifications, and a discussion of limitations. Overall, the paper makes a valuable and practical contribution to AI for Science by addressing scenarios with incomplete physical laws. Despite some limitations in benchmarking, the work is well motivated and demonstrates meaningful advances.